# Fire Protection Performance and Thermal Behavior of Thin Film Intumescent Coating

**Jing Han Beh [1], Ming Chian Yew [2,](IDico), Ming Kun Yew [3] and Lip Huat Saw [2]**

[1]   Department of Architecture and Sustainable Design, Lee Kong Chian Faculty of Engineering Science, University of Tunku Abdul Rahman, Cheras, 43000 Kajang, Malaysia
[2]   Department of Mechanical and Material Engineering, Lee Kong Chian Faculty of Engineering and Science, University of Tunku Abdul Rahman, Cheras, 43000 Kajang, Malaysia
[3]   Department of Civil Engineering, Lee Kong Chian Faculty of Engineering and Science, University of Tunku Abdul Rahman, Cheras, 43000 Kajang, Malaysia
*   Correspondence: yewmc@utar.edu.my or yewmingchian@gmail.com; Tel.: +603-9086-0288

**Abstract:** This paper presents the heat release characteristics, char formation and fire protection performance of thin-film intumescent coatings that integrate eggshell (ES) as an innovative and renewable flame-retardant bio-filler. A cone calorimeter was used to determine the thermal behavior of the samples in the condensed phase in line with the ISO 5660-1 standard. The fire resistance of the coatings was evaluated using a Bunsen burner test to examine the equilibrium temperature and formation of the char layer. The fire propagation test was also conducted according to BS 476: Part 6. On exposure, the samples X, Y, and Z were qualified to be Class 0 materials due to the indexes of fire propagation being below 12. Samples Y and Z reinforced with 3.50 wt.% and 2.50 wt.% of ES bio-filler, respectively, showed a significant improvement in reducing the heat release rate, providing a more uniform and thicker char layer. As a result, the addition of bio-filler content has proven to be efficient in stopping the fire propagation as well as reducing the total heat released and equilibrium temperature of the intumescent coatings.

**Keywords:** acrylic resin; bio-filler; cone calorimeter; heat release rate; intumescent coating; steel

---

## 1. Introduction

According to world fire statistics, there are more than 322,000 people per year suffering in fire accidents due to the ineffectiveness of fire protection systems. Hence it is important to implement an effective fire protection system in every building to protect the occupants and building whenever there is a fire outbreak. Intumescent fire protective materials perform as a passive fire protection system and play a crucial role in fulfilling the fire safety building regulations to effectively stop the advance of a fire. Steel structure starts to lose its mechanical properties (temperature >500 °C) and tends to buckle, leading to failure of the building structures. Indeed, the fire safety rules and regulations in buildings are of paramount importance to ensure the evacuation time and safety for occupants [1].

Intumescent coatings are mainly designed to reduce the heat and fire propagation on the substrate. Interior decorative materials in buildings are mostly combustible products that are a serious hazard in a fire. In general, most of the flame-retardant materials were developed due to the smoke and toxic gas produced from thermal decomposition of the commonly deployed brominated flame-retardant products [2]. Fires are unavoidably used to produce a lot of energy and heat, which might lead to serious injury or death [3–5].

The applications of fire protective coatings are one of the most effective ways to protect different substrates toward a fire. The expansion process of intumescent fire protective coating is due to

---

the physical and chemical interactions of three main flame-retardant additives: (1) Ammonium polyphosphate (APP) acts as an acid source, (2) pentaerythritol (PER) acts as a carbon source and (3) melamine (MEL) acts as an expanding agent. The use of flame-retardant ingredients may prevent a small flame towards a major catastrophe. The polymer binder turns out to be important due to two extensive properties: It contributes to the char layer growth and controls the development of even char foam structure [6,7]. Several advantages of using intumescent fire protective coatings over other approaches of structural fire protection are the artistically attractive appearance it gives to the substrates, fast application, easy to cover complex details and maintaining the intrinsic properties of steel structures [8,9].

This research highlights a renewable chicken eggshell (ES) flame-retardant bio-filler and its important role in industrial coatings. ES waste is an aviculture by-product, which causes a serious conservation risk due to its disposal constitutes. ES waste comprises about 5% organic materials and 95% calcium carbonate in calcite form [10,11]. ES waste can create new value by being converted into profitable products. Its biochemical composition and accessibility make ES a latent source for renewable flame-retardant bio-filler, which improved mechanical and thermal properties of coatings and bio-polymer composites [12–21]. ES also offers benefits for various industrial applications, as it is lightweight, inexpensive, environmentally friendly, has high thermal stability and is available in bulk quantities [22–31]. In this research work, the performance of a steel plate coated with an intumescent coating was tested by using a Bunsen burner. In addition, the fire propagation and fire behavior of intumescent coatings were tested according to BS 476: Part 6 and ISO 5660-1 [32,33], respectively. These thin-film intumescent coatings were evaluated with respect to fire behavior analysis with thermal characteristics in a cone calorimeter and the fire-resistive performance.

## 2. Experimental

### 2.1. Materials

In this experimental work, the three main components of materials used for the preparation of intumescent coatings were (1) the halogen-free flame retardant additives: Ammonium polyphosphate (particle size <15 μm), melamine (particle size <40 μm), pentaerythritol (particle size <40 μm), (2) the flame retardant fillers: Aluminum hydroxide with a specific surface area in a range between 0.5 to 50 m$^2$/g, magnesium hydroxide (specific surface area <8 m$^2$/g), titanium dioxide (particle size <40 μm, specific surface area = 150 m$^2$/g) and eggshell bio-filler (mean particle size = 22.99 μm and specific surface area = 148.41 m$^2$/g) [11] and (3) the polymer binder: Acrylic resin, which has slow-burning or even self-extinguishing behavior when exposed to fire. Moreover, it does not generate harmful smoke or gases. The eggshell powder preparation is shown in Figure 1.

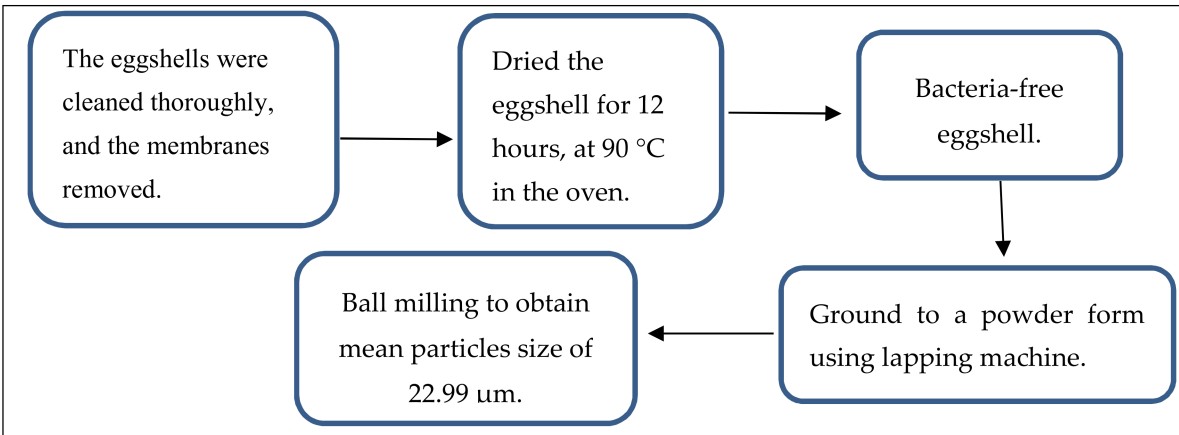

**Figure 1.** Flow chart of chicken eggshell (ES) powder preparation.

Four kinds of thin-film intumescent coatings W, X, Y and Z were prepared and tested to evaluate the fire protection properties and combustion performances of the coatings. The sample details are tabulated in Table 1.

**Table 1.** Specifications of experimental materials.

| Ingredients | Parts by Weight for Formulations | | | |
|---|---|---|---|---|
| | W | X | Y | Z |
| APP | 20 | 20 | 20 | 20 |
| MEL | 10 | 10 | 10 | 10 |
| PER | 10 | 10 | 10 | 10 |
| $TiO_2$ | 5.00 | 3.40 | 3.20 | 3.00 |
| $Al(OH)_3$ | – | 2.80 | 3.30 | 2.75 |
| $Mg(OH)_2$ | 5.00 | 3.80 | – | 1.75 |
| ES | – | – | 3.50 | 2.50 |
| **Polymer Binder** | **W** | **X** | **Y** | **Z** |
| Acrylic resin | 50 | 50 | 50 | 50 |
| Weight * (g) | 26.70 | 24.80 | 25.80 | 25.90 |
| Thickness (mm) | 1.50 | 1.50 | 1.50 | 1.50 |
| Density $(g/cm^3)$ | 1.780 | 1.653 | 1.720 | 1.727 |

* Sample sized for cone calorimeter test = 10 cm (*l*) × 10 cm (*w*) × 1.5 mm (*t*)

### 2.2. Fire Protective Test

The fire protective test allowed the observation of the development of the char layer and the evolution of temperature when exposed to fire to determine the performance of the intumescent fire protective coatings. The intumescent formulation was applied onto a steel plate after being grit-blasted (dimensions of steel plate: $100 \times 100 \times 2.6\text{-mm}^3$) by using a gun sprayer. This method was repeated 3–5 times until a 2.0 ± 0.2 mm coating thickness was attained. The Bunsen burner of the gas tank consumption was about 160 g/h, and the coated steel plate with the intumescent coating was mounted vertically and tested for 60 min of fire (about 1000 °C). In this fire test, 400 °C was chosen as a critical temperature for the coated steel under the small-scale fire test. The time-temperature curves of the coated steel plates were recorded and verified using a model of 307/308 hand-held mini thermometer. The temperature profile and thickness of the char layer on the backside of the steel plates were recorded and evaluated (see Figure 2).

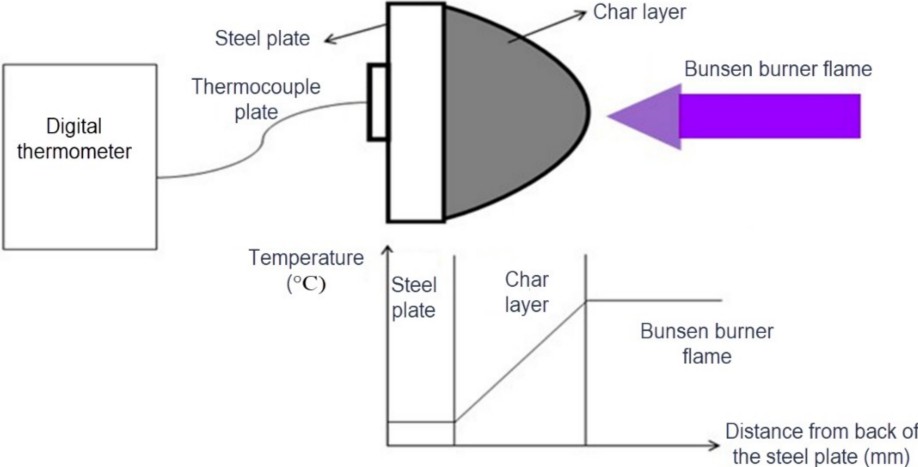

**Figure 2.** Schematic of the experimental setup for fire protective test [31]. Figure 2 is adapted with permission from [31]. Copyright 2018 Elsevier.

### 2.3. BS 476: Part 6 Fire Test

The fire propagation test according to BS 476: Part 6 was conducted and evaluated in this experimental work [32]. The experimental method consisted of exposing the coated steel with an intumescent formulation to a standard small flame for 20 min, with 2 kW of an extra irradiance from the third to the final minute of the fire propagation test. The temperature of the grown ignition intumescent coatings was recorded. This was compared to the temperatures produced from the steel plate coated with the intumescent coating. The result stated was the index of fire propagation, which offers a relative measure of the involvement to the evolution of the fire made by the flat coating surface. The coated steel plate with the intumescent fire protective coating was exposed to the fire conditions. To be Class 0 certified the fire propagation index (*I*) must be below 12. In this fire test, a better fire protective material was determined by a lower numerical value of the index.

In addition, the heat rate and amount of heat grown by the coating sample were measured and considered under prearranged conditions. The steel plates with a standard dimension: $225 \times 225 \times 2.3\text{-mm}^3$ were coated with a thickness of $1.5 \pm 0.1$ mm intumescent coating. Moreover, the index of fire propagation performance was calculated using the below equations:

$$I_1 = \sum_{t=0.5}^{t=3} \frac{\theta_m - \theta_c}{10t} \tag{1}$$

$$I_2 = \sum_{t=4}^{t=10} \frac{\theta_m - \theta_c}{10t} \tag{2}$$

$$I_3 = \sum_{t=12}^{t=20} \frac{\theta_m - \theta_c}{10t} \tag{3}$$

$$I = I_1 + I_2 + I_3 \tag{4}$$

Where:

*I* = index of intumescent coating performance;
*t* = time from the beginning at which readings were taken (min);
$\theta_m$ = temperature of the intumescent coating at time *t*;
$\theta_c$ = temperature of the calibration curve at time *t*.

### 2.4. Sample Preparation for the Cone Calorimeter Test

Before the cone calorimeter test, sample sizes (10 cm × 10 cm × 1.5 mm) were maintained at $50 \pm 5\%$ relative humidity (RH) and $23 \pm 2$ °C. The pretreated coating samples were enveloped with aluminum foil (thickness: 0.03–0.05 mm) with the shiny side of the aluminum foil facing the sample. The coating sample was wrapped without any treatment and the non-exposed surface was covered with foil, which typically forms to the cone calorimeter. One of the most acceptable and internationally recognized fire testing apparatuses is the cone calorimeter. This equipment test was conducted in accordance with the ISO 5660-1 standard. The cone calorimeter is used to examine the fire characteristics of the sample with different measurements simultaneously. The most important parameter to determine a fire's hazard level is to obtain the value of the heat release rate (HRR) of the sample. The specifications of the samples are shown in Table 1.

In this cone calorimeter test, the prepared coating sample and the holder were located on a mass measurement device. All the experiment works were evaluated by employing the coating samples in the same holder under a heater of the cone calorimeter. The fire situation comprised four stages: (1) Ignition, (2) growth, (3) fully developed, and (4) decay. The heat flux of 50 kW/m$^2$ was set and conducted by corresponding to the fully developed fire stage. The distance between the cone and the coating sample was 6 cm. The spark power and the igniter were removed when the ignition or

temporary flame occurred and the time was recorded. If the flame went out after removing the spark power, the igniter was re-inserted within 5 s and then the spark was maintained until test completion. The coating sample and sample holder were detached after collection of all data. Each pretreated coating samples were tested three times according to the standard. The experimental data of the coating sample were calculated based on the average of three tests [33].

Flammability Test

For this flammability test, a heat flux of 50 kW/m$^2$ was irradiated to the square samples and measured in the horizontal position. The following parameters were determined during the test: The heat release rate (HRR), peak HRR, time to ignition (TTI), and total weight loss. The time to fire start on the coating surface due to heat radiation is known as the TTI. The following equations were used to calculate the HRR:

$$\dot{q}''(t) = \frac{q(t)}{A_s} \tag{5}$$

$$\dot{q}(t) = \left(\frac{\Delta h_c}{r_0}\right)(1.10)\, C\, \sqrt{\frac{\Delta P}{T_e}} \frac{\left(X_{O_2}^o - X_{O_2}(t)\right)}{1.105 - 1.5 X_{O_2}(t)} \tag{6}$$

where $\dot{q}''$ is the rate of heat release per unit area (kW/m$^2$), $\dot{q}$ is the HRR (kW), $A_s$. is the initially exposed area (m$^2$), $\Delta h_c$ is the net heat of combustion (kJ/kg), and $r_0$ is the stoichiometric oxygen/fuel mass ratio (-). The maximum intensity of an HRR curve is determined by peak of heat released rate (PHRR).

## 3. Results and Discussion

### 3.1. Bunsen Burner

The temperature profiles and the development of char layers of the steel plates coated with intumescent coating formulations were recorded and compared. The data of time-temperature curves of the coatings are presented in Figure 3. Samples W, X, Y, and Z showed comparable temperature profiles after the test. During the first 10 min of fire, there was no difference in the temperature of all coating samples, and the temperature increased rapidly to 181, 170, 163 and 159 °C for samples W, X, Z, and Y, respectively. After 15 min of fire, the equilibrium temperatures were reached for all coatings and remained almost unchanged until 60 min of fire. The small-scale fire test results demonstrated that the equilibrium temperatures of curves W, X, Y, and Z were 173, 168, 155 and 160 °C, respectively.

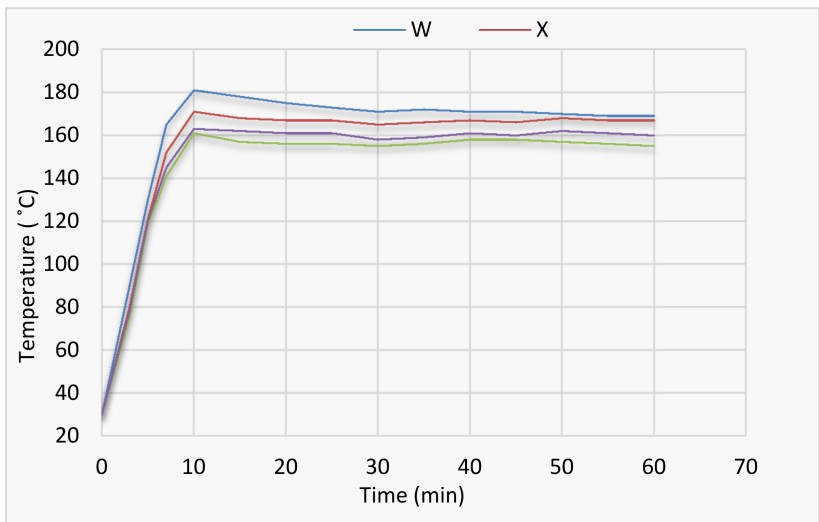

**Figure 3.** The time-temperature curves of the coated steel plates with coating samples.

Moreover, Figure 4 exhibited that the thicknesses and expansion rates for samples Y and Z containing ES bio-filler were 39.50 mm-0.625 mm/min and 37.50 mm-0.59 mm/min, respectively. Sample Y had the best fire protection performance in terms of its equilibrium temperature and char formation compared to samples W, X, and Z. The growth of a multicellular char layer of sample Y was mainly attributed to the decarbonation of 3.5 wt.% of ES. It formed calcium oxide by releasing non-combustible carbon dioxide gas on heating, as follows:

$$CaCO_3 \text{ (s)} \rightarrow CaO \text{ (s)} + CO_2 \text{ (g)} \tag{7}$$

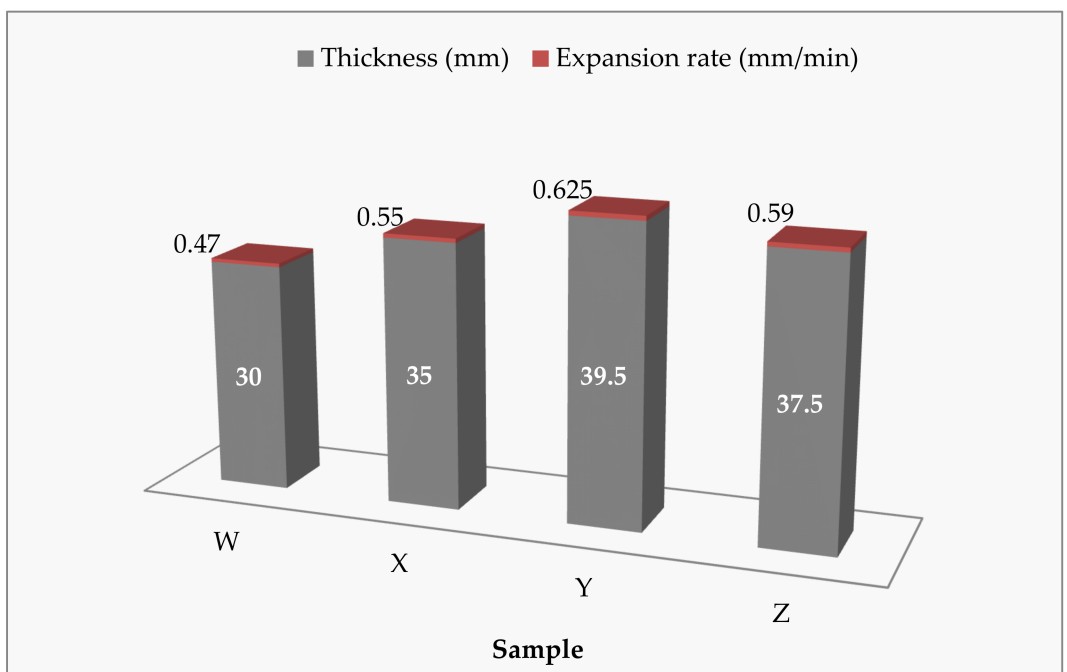

**Figure 4.** The expansion rate and thickness of char layer of intumescent coating samples.

In addition, the expansion of the char layer can be initiated due to physical and chemical reactions contributed by an appropriate mixture of flame-retardant materials and binder, or the development of a cohesive structure during heating. This dense char layer could trap the degradation ingredients into the residue and result in a rounded swelling. This protecting layer declines the heat transfer from the heat source to the underlying steel in maintaining the integrity of the protected substrate against fire. The outcomes demonstrated that the coating comprising phosphate, nitrogen, ES, $TiO_2$, $Mg(OH)_2$, $Al(OH)_3$ containing fire-retardant elements significantly contributed to a better fire protection performance, which resulted from the formation of a uniform and dense char layer. It was found that there was a correlation between the thickness of the char layer and the equilibrium temperature. This shows that the thickness of the char layer affected the fire protection performance of the coating.

### 3.2. BS 476: Part 6

The BS 476: Part 6 fire test found that all samples fulfilled the requirement, except the sample W (($I$) = 22.3). The index and sub-index of the fire propagation test for all coating samples are presented in Table 2.

**Table 2.** The index and sub-index of BS 476: Part 6, fire propagation test.

| Time (min) | Calibration, Temperature (°C) | Coating W (°C) | Coating X (°C) | Coating Y (°C) | Coating Z (°C) |
|---|---|---|---|---|---|
| 0.5 | 14 | 18 | 14 | 12 | 11 |
| 1 | 18 | 21 | 18 | 16 | 15 |
| 1.5 | 23 | 26 | 23 | 19 | 19 |
| 2 | 27 | 30 | 27 | 23 | 22 |
| 2.5 | 30 | 34 | 31 | 26 | 26 |
| 3 | 34 | 38 | 34 | 30 | 29 |
| 4 | 72 | 122 | 55 | 54 | 60 |
| 5 | 108 | 212 | 133 | 129 | 150 |
| 6 | 129 | 274 | 169 | 181 | 179 |
| 7 | 148 | 321 | 213 | 202 | 208 |
| 8 | 166 | 364 | 227 | 219 | 226 |
| 9 | 182 | 378 | 272 | 234 | 236 |
| 10 | 192 | 405 | 290 | 244 | 241 |
| 12 | 214 | 417 | 295 | 249 | 246 |
| 14 | 230 | 418 | 302 | 253 | 250 |
| 16 | 238 | 416 | 304 | 258 | 253 |
| 18 | 246 | 403 | 306 | 260 | 253 |
| 20 | 257 | 385 | 308 | 263 | 259 |
| Sub index 1 ($I_1$) | | 1.6 | 0.2 | 0.1 | 0 |
| Sub index 2 ($I_2$) | | 15 | 5.1 | 4.6 | 4.4 |
| Sub index 3 ($I_3$) | | 5.7 | 1.5 | 1.2 | 0.7 |
| Index of Performance ($I$) | | 22.3 | 6.8 | 5.9 | 5.1 |

The BS 476: Part 6 test results showed that the sub-index ($I_1$:$I_2$:$I_3$) of coating samples W, X, Y and Z was (1.6:15:5.7), (0.2:5.1:1.5), (0.1:4.6:1.2) and (0:4.4:0.7), respectively. The index ($I$) results for the same coating samples were 22.3, 6.8, 5.9 and 5.1, respectively. It emphasized that the sub-index must be below 6 and the index of fire propagation must be below 12 for the coating samples to be certified as Class 0 materials. Among all coating samples, only sample W did not qualify as a Class 0 material since its index was 22.3 ($I$), which is out of the index of performance of this category ($I > 12$).

Evaluation of the fire propagation index for samples W, X, Y and Z revealed that samples Y (($I$) = 5.9) and Z (($I$) = 5.1) with 3.5 wt.% and 2.5 wt.% of ES, respectively, showed a great reduction in fire propagation index compared to sample W. It can be concluded that the incorporation of ES bio-filler into the coating formulation led to substantial inhibition of fire propagation, which could be contributed to the decarbonation of calcium carbonate at a high decomposition temperature [28].

In addition, sample X (($I$) = 6.8) also exhibited a significant improvement in the reduction of fire propagation compared to sample W. This coating formulation showed an appropriate combination of $TiO_2$/$Al(OH)_3$/$Mg(OH)_2$ flame retardant fillers with flame retardant ingredients led to a significant improvement in stopping the fire propagation behavior. This phenomenon is due to the main phosphorus element of ammonium polyphosphate (APP), which could easily respond with different oxides during a fire to produce ceramic-like solid materials (*X*-O-P species, *X* = Ti, B, Al, Mg, etc.). This develops a more cohesive and dense char structure [1,34,35]. The properties of the char structure are associated with fire protection performance of the sample [36,37].

### 3.3. Cone Calorimeter Test

The results of the cone calorimeter test using 50 kW/m$^2$ heat fluxes are shown in Table 3. The overall burning time of all intumescent coating samples was about 700 to over 900 s, and the TTI was 8–10 s.

**Table 3.** Data of the cone calorimeter test of samples.

| Sample | Peak of Heat Released Rate (kW/m$^2$) | Total Heat Released (MJ/m$^2$) | Thickness of Char Layer (mm) | Time to Ignition (s) | Residual Weight (wt.%) |
|--------|---------|---------|---------|---------|---------|
| **W** | 106.03 | 22.4 | 21.0 | 9 | 43.85 |
| **X** | 111.86 | 21.6 | 30.0 | 8 | 46.12 |
| **Y** | 91.00 | 11.5 | 35.5 | 10 | 61.81 |
| **Z** | 99.98 | 12.0 | 34.0 | 10 | 58.48 |

According to cone measurements, the TTI values of samples W, X, Y, and Z were 9, 8, 10 and 10, respectively. The TTI of the high-density samples Y and Z, which contained ES bio-filler, had a longer time than those of the lower density sample X, demonstrating that the main factors were the density and decomposition temperature of the flame-retardant fillers [38]. In addition, the remaining mass of coating samples W, X, Y, and Z were 43.85%, 46.12%, 61.81% and 58.48%, respectively, after the test.

Therefore, samples Y and Z incorporated with ES were difficult to ignite and contribution to the TTI value and residual weight compared to samples W and X, due to its higher decomposition temperature. It is important to examine the profile of the HRR curve over time as it may reveal evidence on the varying thermal behavior of the heating process due to physical and chemical reactions of intumescent coatings.

Figure 5 displays the HRR versus time profiles of the coating samples after ignition. The burning behavior of the entire samples exhibited a single peak. Sample X showed the maximum peak of 111.86 kW/m$^2$ at 35 s, which was higher than other maximum peaks of samples W, Y and Z of 106.03, 91.00 and 99.98 kW/m$^2$, respectively. The PHRR of sample Y was the lowest among all the samples due to its positive synergistic effect in reducing the heat release rate with the addition of 3.5 wt.% ES bio-filler into flame retardant ingredients and binder. The results show that the HRR value with the incorporation of bio-filler of samples Y and Z maintained at a level below 20 kW/m$^2$ in the time range between 250–700 s ignition, while the HRR of samples W and X without addition of ES bio-filler decreased slowly and maintained at a level below 20 kW/m$^2$ after at 600 s ignition.

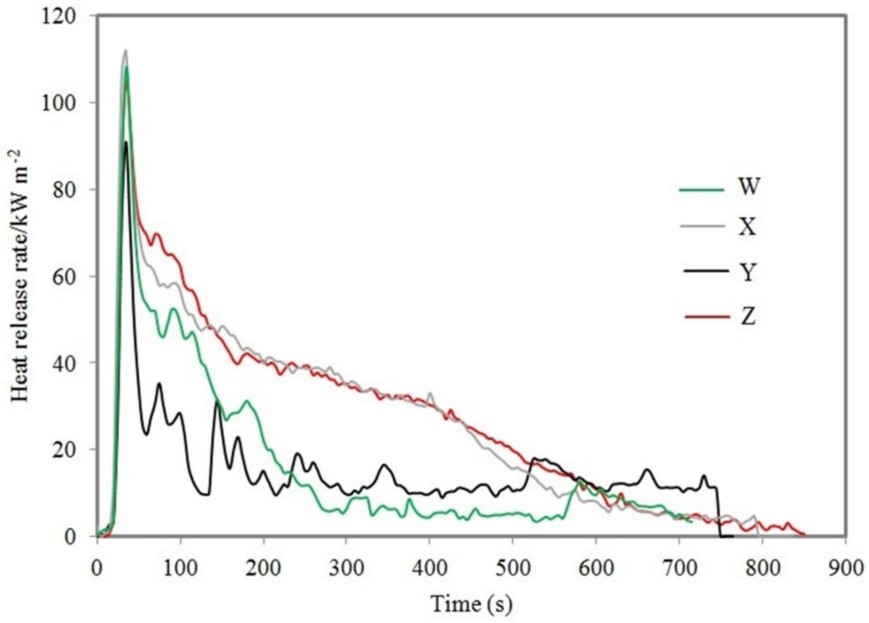

**Figure 5.** Heat release rate (HRR) of the samples.

Figure 6 displays the curves of total heat released (THR) against the time of the intumescent coating samples. Samples W and X showed higher values of 22.4 and 21.6 MJ/m$^2$, respectively, compared to

samples B and D, which had lower values of 11.5 and 12.0 MJ/m². The THR of samples W and X rose sharply and tended to follow a smooth curve after ignition. Samples Y and Z showed a very significant improvement in the reduction of the THR with the addition of the novel ES bio-filler. This result indicated that the heat release of samples Y and Z during combustion was very small and not enough to sustain combustion without external heat flux. The excellent flame retardancy properties of samples Y and Z were probably caused by the existence of the carboxylic group and calcium ions in the calcium carbonate and the carbon source, which promoted a dehydration reaction and decarboxylation reaction to release non-burning gases, such as $H_2O$ and $CO_2$.

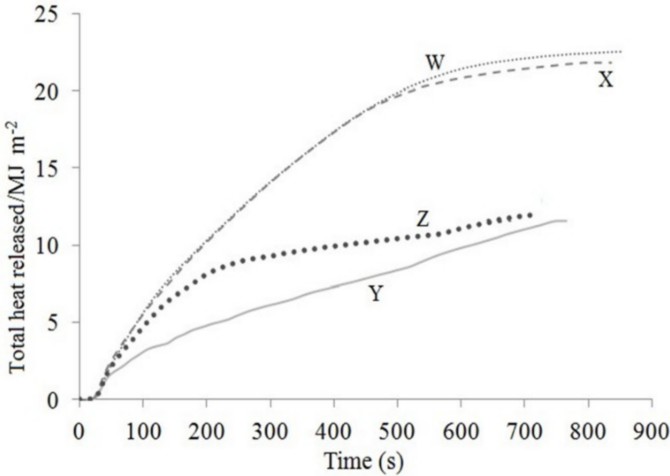

**Figure 6.** The curves of total heat released (THR) versus time of the samples.

The coating samples before and after the cone calorimeter test are shown in Figure 7a–d. Samples Y and Z, which comprise ES, had more effective char formation and expansion rate compared to samples W and X, due to appropriate combinations of flame retardant fillers (Y-ES/Al(OH)$_3$/TiO$_2$) and Z-ES/Al(OH)$_3$/Mg(OH)$_2$/TiO$_2$). This could be attributed to the physical and chemical integration of the flame-retardant ingredients. The decomposition of $Mg(OH)_2$ and $Al(OH)_3$ flame-retardant fillers is described in the equations below:

$$Mg(OH)_2 \ (s) \rightarrow MgO \ (s) + H_2O \ (g) \tag{8}$$

$$2Al(OH)_3 \ (s) \rightarrow Al_2O_3 \ (s) + 3H_2O \ (g) \tag{9}$$

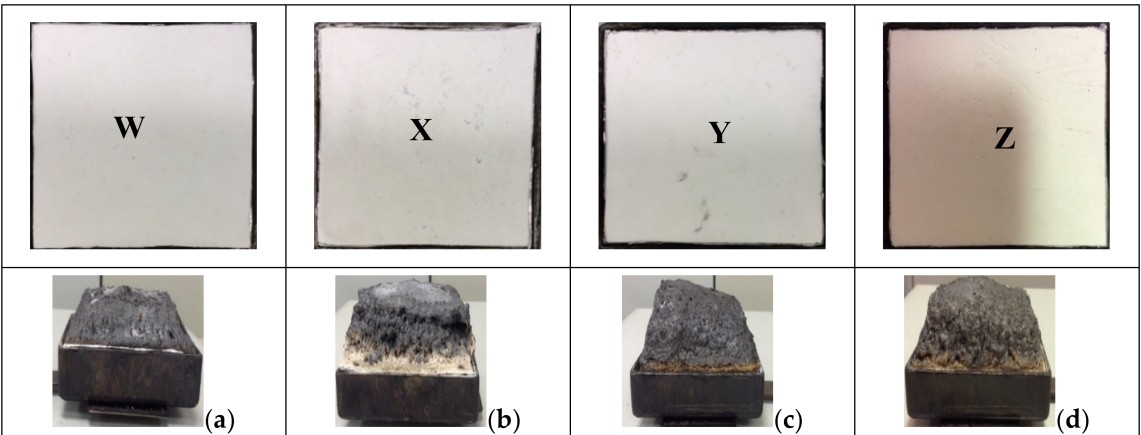

**Figure 7.** The coating samples before (W, X, Y and Z) and after (**a–d**) the cone calorimeter test.

The properties of the $Al(OH)_3$ flame-retardant filler displays strong reversibility of the dehydration reaction when exposed to heat, resulting in good fire resistance performance, since water released inside the particles recombine with the reactive surface of the freshly formed alumina [39]. However, the endothermic decomposition of the $Mg(OH)_2$ filler would attribute to a gaseous water phase, which could enclose the flame by eliminating oxygen and dilute combustible gases by reducing the total heat released [40].

Sample Y revealed the highest rate growth and thickest char layer among the coating samples, as presented in Figure 7c. The development of the multicellular layer could have been originated by the release of non-combustible $CO_2$ due to the decarbonation of ES bio-filler, which induces swelling by trapping the degradation products into the residue, as explained in Section 3.1.

The thermal degradation of ammonium polyphosphate can easily react with flame-retardant fillers to form a ceramic-like material, which increases the char formation by giving a dense and uniform char layer, which could insulate and protect the unprotected substrate in a fire [34,35].

## 4. Conclusions

The thermal characteristics of four intumescent coating formulations have been studied in accordance with the BS 476 Part 6: Fire propagation test and ISO 5660-1 cone calorimeter standard test under atmospheric conditions with a piloted ignition. The incorporation of the ES bio-filler in the intumescent formulation led to a good thermal resistance and fire protection performance. It was found that all the parameters that characterize coating thermal resistance, such as TTI, HRR, and THR, decreased when 3.50 wt.% and 2.50 wt.% ES bio-filler was added to samples Y and Z. Hence, this study revealed that the addition of ES bio-filler strongly influenced the thermal properties and formation of the char layer of intumescent coatings. The coated samples X, Y and Z showed neither fire propagation nor afterglow combustion. Appropriate combinations of $Al(OH)_3/TiO_2/ES$ in the coating formulation decreased the index value of fire propagation and HRR, whilst providing a thicker and more uniform char layer. The addition of renewable ES bio-filler showed significant enhancement in fire protection and the quality of the intumescent fire protective coatings, as well as being beneficial to the environment. In general, it can be determined that intumescent coatings display significant fire protection qualities in a practical and effective fire protective coating for steel, as shown by the findings of this study.

**Author Contributions:** Interpretation, J.H.B. and M.C.Y.; methodology, J.H.B.; validation, J.H.B., M.C.Y., M.K.Y. and L.H.S.; formal analysis, J.H.B.; investigation, J.H.B. and M.C.Y.; resources, J.H.B.; data collection, J.H.B.; writing—original draft preparation, J.H.B.; writing—review and editing, M.C.Y.; visualization, M.K.Y. and L.H.S.; supervision, M.C.Y.; project administration, M.C.Y.; funding acquisition, M.C.Y.

**Funding:** This project was funded by the University of Tunku Abdul Rahman under the UTARRF.

**Acknowledgments:** The authors would like to express their sincere gratitude to City University of Hong Kong for providing the laboratory services.

**Conflicts of Interest:** There is no conflict of interest declared by authors.

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
