# Peer review of "Fire Protection Performance and Thermal Behavior of Thin Film Intumescent Coating"

_coatings, doi:10.3390/coatings9080483_

Reviewer 1 Report

The Authors presented interesting findings in field of designing intumescent coatings.

However, the paper needs some improvments.

I have the following comments regarding this article:

·         Please change “The fire resistive of” into “The fire resistance of” (line 17)

·         Try to rephrase “incompetence of fire protection system”, I think the word “incompetence” is not suitable in this context (line 29)

·         Try to rephrase “may avoid a small fire from”, I think the word “avoid” is not suitable in this context (line 47)

·         Please explain the abbreviations APP, MEL, PER in Table 1

·         According to Table 1 the samples thickness is 1.5 mm but according to the text (line 83) it is 2.0 ± 0.2 mm or 1.5 ± 0.1 mm (line 111) which value is true?

·         If you want to show both height of char (thickness) and expansion rate at the same Figure please use logarithmic scale – Figure 3.

·         Please change “Figure 3.0 Height” into “Figure 3 Height” (line 191)

·         Please change “Figure 1.0 Schematic” into “Figure 1 Schematic” (line 94)

·         Please change “Figure 4.0” into “Figure 4” (line 240)

·         Not all the equations are numbered – lines 118, between 157 and 158,

·         Please divide Figure 4.0 into a Figure and a Table or only a Table will be enough – the data are repeated.

·         Add units to Table 2 - °C?

·         To my knowledge the THR abbreviation stands for Total Heat Released not Total Heat Rate.

·         Please characterize the materials used for the intumescent coating preparation: e. g. particle size, specific surface area of APP, MEL, PER, TiO2, Al2(OH)3, Mg(OH)2. How the egg shells were prepared? What is the characteristic of the acrylic resin?

·         The data from Table 2 could be presented in more comprehensive form – column graph?

Author Response

Reviewers’ comments: 

(1) Please change “The fire resistive of” into “The fire resistance of” (line 17)

Response: “The fire resistive of” has been revised into “The fire resistance of” accordingly (line 17). Thank you.

(2) Try to rephrase “incompetence of fire protection system”, I think the word “incompetence” is not suitable in this context (line 29)

Response: “incompetence of fire protection system” has been rephrased to “ineffectiveness of fire protection system” accordingly (line 29). Thank you.

(3) Try to rephrase “may avoid a small fire from”, I think the word “avoid” is not suitable in this context (line 47)

Response: “may avoid a small fire from” has been rephrased to “may prevent a small fire from” accordingly in the revised manuscript (line 49). Thank you.

(4) Please explain the abbreviations APP, MEL, PER in Table 1

Response: APP stands for Ammonium Polyphosphate, MEL stands for Melamine and PER stands for Pentaerythritol. The abbreviations APP, MEL and PER have been added accordingly in the introduction of the revised manuscript (line 48). Thank you.

(5) According to Table 1 the samples thickness is 1.5 mm but according to the text (line 83) it is 2.0 ± 0.2 mm or 1.5 ± 0.1 mm (line 111) which value is true?

Response: The sample thickness of 1.5 mm is for cone calorimeter test according to Table 1, sample sized for cone calorimeter test = 100 mm (w) x 100 mm (l) x 1.5 mm (t) has been added accordingly in the revised manuscript (line 77). Thank you.

(6) If you want to show both height of char (thickness) and expansion rate at the same Figure please use logarithmic scale – Figure 3.

Response: The chart type has been changed to stacked column in 3-D to better describe the char (thickness) and expansion rate of samples. Thank you.

(7)  Please change “Figure 3.0 Height” into “Figure 3 Height” (line 191)

·         Please change “Figure 1.0 Schematic” into “Figure 1 Schematic” (line 94)

·         Please change “Figure 4.0” into “Figure 4” (line 240)

·         Not all the equations are numbered – lines 118, between 157 and 158,

Response: All Figures 1.0, 3.0 and 4.0 have been revised to Figure 1, Figure 3 and Figure 4 accordingly. All the equations are numbered accordingly in the revised manuscript. Thank you.

(8) Please divide Figure 4.0 into a Figure and a Table or only a Table will be enough – the data are repeated.

Response: Only a Table 3 shows in the revised manuscript (line 255) and the Figure 4 is removed accordingly as suggested. Thank you.

(9) Add units to Table 2 - °C?

Response: The units (°C) have been added into Table 2 (line 229). Thank you.

(10) To my knowledge the THR abbreviation stands for Total Heat Released not Total Heat Rate.

Response: Sorry for the mistake. Total Heat Released (THR) has been added in the revised manuscript (line 282). Thank you.

(11) Please characterize the materials used for the intumescent coating preparation: e. g. particle size, specific surface area.

Response: In this experimental work, three main components of materials used for the preparation of intumescent coatings are (1) halogen-free flame retardant additives: ammonium polyphosphate (particle size < 15 μm), melamine (particle size < 40 μm), pentaerythritol (particle size < 40 μm), (2) flame retardant fillers: aluminum hydroxide (specific surface area in a range of between 0.5-50 m2/g), magnesium hydroxide (specific surface area < 8 m2/g), titanium dioxide (specific surface area = 150 m2/g; particle size < 40 μm) and eggshell bio-filler ( mean particle size = 22.99 μm and specific surface area = 148.41 m2/g) [11] and (3) polymer binder: acrylic resin, which has slow burning or even self-extinguishing behavior when exposed to fire. Moreover, it does not generate harmful smoke or gases (line 72-80).

(12) How the egg shells were prepared?

Response: Figure 1 Flow chart of chicken eggshell powder preparation has been added accordingly in the revised manuscript. (1) The eggshells were cleaned thoroughly; and the membranes removed. (2) Dried the eggshell for 12 hours, at 90 °C in the oven. (3) Bacteria-free eggshell and (4) Ground to a powder form using lapping machine to obtain mean particles size of 22.99 μm (line 81-82).

(13) What is the characteristic of the acrylic resin?

Response: Acrylic resin has slow burning or even self-extinguishing behavior when exposed to fire. Moreover, it does not generate harmful smoke or gases (line 78). Thank you.

(14) The data from Table 2 could be presented in more comprehensive form – column graph?

Response: Table 2 is a standard form to present the sub index and index of performance of the samples under the BS476 Part 6: Fire Propagation. Thank you.

Reviewer 2 Report

The paper needs some revesion in light of the following points:

Alternative flame retardants – in general – were developed due to the toxic products produced from thermal decomposition of the commonly deployed brominated flame retardant (see for example Progress in Energy and Combustion Science 2019, 70, 212-259). This should be mentioned in the introduction.

Thickness and Expansion rate should be plotted in two separate figures. Likewise, please improve quality of Figure 2, you may use Origin software. The quality of Figures overall is rather poor. 

Would it be possible to describe chemical events that accompany heat release for formation of residue?

How HR values compare with other similar materials? Please elaborate more into this.

Author Response

Reviewers’ comments:

(1) Alternative flame retardants – in general – were developed due to the toxic products produced from thermal decomposition of the commonly deployed brominated flame retardant (see for example Progress in Energy and Combustion Science 2019, 70, 212-259). This should be mentioned in the introduction.

Response: The article has been mentioned in the introduction (line 40) and cited in the order reference number [2]. Thank you.

(2) Thickness and Expansion rate should be plotted in two separate figures. Likewise, please improve quality of Figure 2, you may use Origin software. The quality of Figures overall is rather poor. 

Response: The thickness and expansion rate have been changed to stacked column in 3-D to better describe the char layer formation of the intumescent coatings when exposed to fire. Thank you.

(3) Would it be possible to describe chemical events that accompany heat release for formation of residue?

Response: Thank you for the suggestion. Chemical events that accompany heat release for formation of residue for each coating formulations will be examined in the next paper. In this research project, authors are mainly focused on chemical events of the influences of flame-retardant fillers for intumescent coatings when exposed to heat as follows:

CaCO3 (s) à CaO (s) + CO2 (g)

2Al(OH)3 (s) → Al2O3 (s) + 3H2O (g)

Mg(OH)2 (s) → MgO (s) + H2O (g)

However, the degradation products of flame-retardant additives can easily react with flame-retardant fillers contain oxides (Mg(OH)2, TiO2, Al(OH)3, etc) during burning to yield a ceramic-like material, which enhances the char structure formation by giving a stronger and more cohesive char layer, which could isolate the steel substrate from fire and provide better fire protection have been analyzed and explained.

(4) How HR values compare with other similar materials? Please elaborate more into this.

Response: In this research work, the different of the intumescent coating formulations are the contents of the flame-retardant fillers. There are > 93 wt% of each coating formulation has the same content of flame-retardant ingredients (such as flame-retardant additives, polymer binder and flame-retardant fillers). This research work is mainly to examine the effect of the addition of ES content on the HR values compare with other coating formulations without the addition of ES bio-filler.

Round  2

Reviewer 2 Report

Authors have addressed my comments hence I now recommend to accept the manuscript

Author Response

Thank you !